# Raising Awareness on the Clinical and Social Relevance of Adequate Chronic Pain Care

**DOI:** 10.3390/ijerph20010551

**Published:** 2022-12-29

**Authors:** Silvia Natoli, Alessandro Vittori, Marco Cascella, Massimo Innamorato, Gabriele Finco, Antonino Giarratano, Franco Marinangeli, Arturo Cuomo

**Affiliations:** 1Department of Clinical Science and Translational Medicine, University of Rome Tor Vergata, 00133 Roma, Italy; 2IRCCS Maugeri, 27100 Pavia, Italy; 3Department of Anesthesia and Critical Care, ARCO Roma, Ospedale Pediatrico Bambino Gesù, IRCCS, 00165 Rome, Italy; 4Department of Anesthesia and Critical Care, Istituto Nazionale Tumori-IRCCS, Fondazione Pascale, 80131 Naples, Italy; 5Department of Neuroscience, Pain Unit, Santa Maria delle Croci Hospital, AUSL Romagna, 48121 Ravenna, Italy; 6Intensive Care Unit, Azienda Ospedaliero Universitaria Cagliari, 09042 Monserrato, Italy; 7Department of Medical Sciences and Public Health, University of Cagliari, 09042 Monserrato, Italy; 8Department of Surgical, Oncological, and Oral Science (Di.Chir.On.S.), University of Palermo, 90133 Palermo, Italy; 9Department of Anesthesia, Intensive Care and Emergency, Policlinico Paolo Giaccone, 90127 Palermo, Italy; 10Department of Anesthesiology, Intensive Care and Pain Treatment, University of L’Aquila, 67100 L’Aquila, Italy

**Keywords:** chronic pain, pain management, pain therapy network, care pathways, right enforceability

## Abstract

Appropriate pain care should be regarded as a right and effectively guaranteed to people with chronic pain (CP). Law 38, enacted in Italy in 2010, establishes the citizen’s right not to suffer. Twelve years later, such right appears still disregarded in Italy and the current access to adequate pain care reveals significant shortcomings. In addition, a mismatch between CP-associated burden and the available healthcare resources in the framework of our national health system has been observed. This article gathers the perspectives of a Board of Italian anesthesiologists on the state of the art of CP management in Italy and aims at strengthening the scientific rationale and clinical relevance of pursuing the enforceability of the right not to suffer and at promoting widespread multidisciplinary care of patients with CP.

## 1. Introduction

Regarded for a long time as a non-life-threatening condition and often overlooked, pain has historically been under-treated and the clinical sequelae of its poor control disregarded [1,2]. However, recent evidence highlighted a worrisome prevalence of pain worldwide with variability across countries ranging from 9.9% to 50.3% thus endorsing pain as a worldwide primary health issue [3]. Chronic pain (CP) inflicts the greatest loss of productivity than any disease condition [4] and contributes to significant healthcare costs [5,6] Long-lasting untreated or inadequately controlled pain is a chronic condition that severely impacts patients’ quality of life (QoL) through wide-ranging negative physical, psychological, and functional effects [7,8]. Therefore, access to adequate pain care should stand as a non-postponable right [9,10], and the rights of people with CP must be championed [11].

An important milestone in the European healthcare regulatory framework was represented by Law 38, enacted in Italy in 2010, which ratified the citizen’s right not to suffer [12]. The law pioneered recommendations to develop dedicated health centers for palliative care and pain therapies and to provide both inpatient and outpatient care settings to ensure continuity in the diagnostic-therapeutic journey of patients with CP [12]. To date, alongside Italy, few European governments have recognized pain management as a legal obligation [13] and few national health organizations approved a charter of rights for people living with CP [14].

Despite the regulatory framework defined by Law 38/2010, the provision of adequate CP care appears challenging in our country. It is currently missing a multidisciplinary pain management approach stemming from the observation that CP pathology has intricately related biological and psychosocial components. Therefore, the usual treatment delivered by a single department/specialist may not be sufficient for people with CP thus requiring multi-professional teams. Furthermore, there is limited awareness of the opportunities Law 38/2010 can offer to citizens with seven out of ten citizens knowing neither the Law nor the rights that it establishes to avoid the patient “unnecessary suffering [15]. As a result, Italy is the third European country in terms of CP prevalence [16] with about one in four subjects suffering from it and one in three pain patients undiagnosed or referred late to pain therapy centers [17,18,19]. Overall, the missed or partial enforceability of the right not to suffer demands a call to action to mitigate the CP burden, and a greater awareness of rights can no longer be disregarded among the general population and within the medical community. In the last few years, several initiatives intended to both assess the degree of implementation of Law 38/2010 at a national level and bridge the clinical, organizational, and cultural gaps that hamper CP patients’ access to the care they are eligible for were organized [20,21,22]. Recently, a Social Manifesto against pain, spearheaded by the Italian Society of Anesthesia, Analgesia, Resuscitation, and Intensive Care (SIAARTI) and signed by seventeen scientific societies and patient associations, has been issued to foster awareness of the burdensome impact of CP at both individual and societal level [23].

Inspired by the challenges and opportunities to improve pain care in our country, we did reflect on the state of the art of CP management with the aim of advancing pain care towards integrated patterns of care where all the professionals involved in CP patients’ management can convey to develop fully multidisciplinary pain management by efficiently translating research into practice. Following up the premises and auspices of the Social Manifesto against pain, [23] our perspective aims at complementing the current debate, ongoing in several countries at the global level, [24,25] about the need of promoting multidisciplinary pain care, education, advocacy, and research to improve function and QoL for people with CP [24].

## 2. Pursuing Effective Chronic Pain Care: From Law 38/2010 to the Italian Manifesto against Pain

Pursuing the enforceability of the right not to suffer requires the promotion of medical, cultural, and social changes that can make pain management a core component of good clinical practice [26,27]. Therefore, strategies and health and social policy care are needed to mitigate the burdensome consequences of inadequate CP care. Of note, similarly to what has been observed in other countries [28], it has been reported a significant variability in pain management in our country with patients often bouncing around from one health center to another while still suffering from inadequately relieving symptoms [19]. This challenging scenario may stem from two essential shortcomings. The former is the still common attitude, among the general population, to not perceive pain as a disease in itself; therefore, patients are often looking for a specialist caring for that specific disease rather than referring to a pain specialist. The latter is the limited recognition of pain medicine as a medical specialty and the poor awareness, among both patients and clinicians, of the presence of specialized pain centers in both hospital and ambulatory settings.

Law 38/2010 has been groundbreaking in Europe to address the disregarded rights of people eligible for CP and palliative care (Table 1) and over the last few years, several appraisals of the degree of Law 38/2010 implementation and the related achievements have been performed [18,20,22,29]. Overall, as suggested by Marinangeli et al., Law 38/2010 can be regarded as the “magnificent unfinished”: optimal and well organized in its essence but hard to make it real to effectively serve patients with CP [22].

To foster greater awareness of the pitfalls of current CP care and therefore promote better implementation of Law 38/2010, SIAARTI did spearhead a Social Manifesto against pain [23]. Such manifesto stemmed from a three-fold premise: (a) the recognition of the advances in the regulatory framework underlying the Law 38/2010 implementation such as the introduction of coding 96 to recognize pain management as a discipline at a national level [30] and the agreement signed by the State-Regions Conference about the accreditation procedure for the pain therapy networks [31]; (b) greater appreciation of the pain as a biopsychosocial issue and of the clinical implications of the latest edition of the International Classification of Diseases (ICD-11) [1,4]; (c) the insights provided by an extensive survey involving the majority of specialist pain centers within our national pain therapy network [21] (Table 2).

Despite during the last decade several initiatives aimed at translating the guiding principles of Law 38/2010 into actionable interventions and promoting disease awareness within the general population to improve pain care in the daily practice of healthcare delivery have been put in place, the way forward is not without shortcomings.

## 3. Current Gaps Hindering Appropriate Care for Patients with Chronic Pain

With one in three undiagnosed pain patients and frequent late patient referral to pain therapy centers, access to pain care reveals burdensome shortcomings in Italy [17,18,32,33]. Now, more than ever, the translation of pain knowledge into practice, as advocated by the International Association for the Study of Pain (IASP) 2022 Global Year [34], is imperative to maximize the opportunities that full implementation of Law 38/2010 could provide. Major diagnostic, therapeutic, and organizational gaps which currently hinder the full enforceability of the right not to suffer and appropriate multidisciplinary CP care are identified.

### 3.1. Poor Disease Awareness and Late Diagnosis

Pain is still regarded by the patients as an inevitable component of the disease thus favoring a frequent “wait and see” attitude on the part of the patients who may first attempt to control the pain by self-medication or simply by underestimating the problem [19]. Patients seek doctor consultations and/or pain center visit months or years after pain occurrence; nonetheless, a high percentage of patients who wait before contacting a pain therapy center are often unaware of its existence. In line with this, a recent cross-sectional study conducted on over 1000 patients with CP reported that one in two patients had suffered from CP for over four years [32]. To this end, it has been suggested that the acknowledgment of chronic and recurrent pain within the recent ICD-11 classification could be a premise for promoting greater consciousness as well as enhanced engagement of both healthcare providers and patients [35]. Advances in pain knowledge have laid the foundation for the ICD-11 classification and supported the notion that CP should be regarded as a disease, rather than a symptom, which carries both meaningful distress and functional impairment [1,36]. It has been suggested that the new ICD-11 CP classification could make CP more “visible” and better inform clinical practice and resource allocation [37]. Overall, coding CP with the ICD-11 may be of help primarily in terms of obtaining a timely diagnosis; therefore, a wider implementation of ICD-11 classification may thus shorten the time to diagnosis. Considering that ICD-9 is the version of ICD currently envisaged in our country by the legislation for the coding of diseases and related problems [38], it is paramount to promote the transition from ICD-9 to the ICD-11 that has come into force on 1 January 2022.

Mounting evidence suggests that most physicians may feel not be adequately prepared to manage patients with CP [39,40,41]. The over-burdened primary care providers, while being at the forefront of pain care delivery pathways, often lack the time and resources to effectively assess and manage CP thus very often choosing to refer patients to specialized pain centers. Accordingly, 75% of patients visiting the Italian pain therapy network are referred by GPs [21], after an unsuccessful trial of pharmacological and non-pharmacological options. Overall, after providing the first level of pain care, GPs should promptly refer CP patients to specialized centers where specialists can offer integrated, expert assessment and management of pain within the context of a multidisciplinary team.

### 3.2. Inadequate Management

CP demands individualized management [42,43]. Therefore, the evaluation of analgesic treatment needs to reflect at a minimum improvement in pain degree of severity and pain-related distress, as well as lower interference with daily life. Currently, clinicians are still not fully aware that pain therapy success should go beyond reducing pain intensity thus not easily shifting toward an emphasis on patient functions. In line with this, it has been reported that although clinicians acknowledge a restoration of functionality as relevant as a significantly lower pain degree, they report limited use of multidimensional questionnaires [21]. Difficulties can be encountered in choosing the most promising therapeutic strategy. A recent survey among Italian GPs and specialists reported that only 1 in 2 physicians report the presence of treatment protocols and management patterns addressing patient pain according to pain type [33].

CP should be addressed by combining both pharmacological and nonpharmacological approaches taking into consideration multiple aspects including its intensity and duration, pathophysiology, symptoms’ complexity, and the coexistence of comorbidities [44,45,46]. Therefore, combining analgesics with different mechanisms of action or pharmacological approaches with invasive techniques may all serve as promising approaches to attain multimodal CP management with the final aim of improving function, and quality of life and facilitating and enabling the return to work [47]. Of note, multimodal pain management has become a fundamental part of perioperative care and may prevent the development of chronic postoperative pain [48,49]. Similarly, optimal treatment of CP (including rheumatic [50], low back pain [51], and osteoarthritis [52,53]) can be achieved by taking advantage of a multimodal approach encompassing structured interdisciplinary programs aimed at providing a multidisciplinary treatment plan. To this end, multimodal CP management should include medications, physical and occupational therapy, rehabilitation, behavioral therapy, mini-invasive and invasive procedures, and so on as illustrated in the analgesic trolley model for pain management proposed by Cuomo et al. [46].

The appropriateness of care is a priority when managing prevalent diseases which need long-term treatment, such as CP [54] To this end, Law 38/2010 aimed at simplifying the procedures to access pain drugs by modifying the Unique Text of Law regarding the use of narcotic and psychotropic substances, thus allowing general practitioners (GPs) to prescribe non-injectable opioids upon adequate training on their use. Although in our country the average per capita year of morphine equivalent dose is much lower than that reported in Northern Europe and the USA, avoiding misuse while guaranteeing all patients with a pain treatment has been regarded as a primary health matter [55]. In Italy, it has been recently reported both a lower consumption of weak opioids and marked heterogeneity in strong opioid consumption across regions [56]. Such findings highlight the achievements and pitfalls of Law 38/2010: The former is greater appropriateness of the use of such drugs for CP patients, as suggested by the tight surveillance of opioid use achieved in our country [57] and the latter is the need for harmonizing the access to pain care according to the equality principles of the law. Nevertheless, standardized protocols for opioid titration as well as for the management of opioid abuse are not currently available in many pain centers [20]. 

Finally, despite Law 38/2010, the possibility to ensure, via the national healthcare system, multidisciplinary care of people with CP is still challenging. To date, the essential levels of assistance (LEA), i.e., the services and benefits that the national health service is required to provide to all citizens, currently exclude chronic pain care from both psychological and rehabilitation settings.

### 3.3. Inhomogeneous Pain Care Delivery

The main objective of Law 38 was to ensure continued patterns of care for patients established through the hospital-primary medicine network. However, only 32.6% of the pain facilities guaranteed a homogeneous care continuity within the network via the use of the electronic medical record (EMR) of patients [20]. In addition, a comprehensive mapping of the Italian pain therapy centers unveiled significant organizational issues that might have contributed to the inhomogeneity of care delivery across regions. Nonetheless, it has been reported that inadequate pain assessment may stem from the limited time physicians have to devote to pain patients’ consultation, the absence of streamlining available tools as well as the scarce proportion of physicians who devote their clinical activity to pain medicine [21].

It has been suggested that clinical care pathways could serve as useful tools to improve the quality of healthcare by facilitating the translation of evidence into practice [58]. Therefore, great efforts have been put in place by multiple scientific bodies and societies to develop best practice recommendations and propose pain care pathways. Despite the presence of multiple scientific bodies involved in pain care including the Italian Society of Emergency Medicine (SIMEU), Italian Resuscitation Council (IRC), Italian Society of Anesthesia, Resuscitation, Emergency and Pain (SIARED), Italian Society 118 System (SIS 118) and Italian Association for the Study of Pain (AISD), there is limited evidence of intersociety recommendations on pain management [59] thus hindering an integrated multidisciplinary pain care delivery. It has been suggested that multiple factors may contribute to the limited adoption of shared clinical pathways including skepticism, difficulty in portraying patients’ clinical picture within a set pathway, and lack of support in obtaining knowledge [60]. In addition, there is limited evidence of shared clinical pathways as most clinicians would perceive them as a limitation (or even loss) to their autonomy in clinical decision-making. As a result, integrated care pathways for pain management in our country are difficult to implement and, if they exist, are limited to very local realities. To this end, it would be desirable to engage the representatives of the main healthcare professional categories daily involved in pain care (not only anesthesiologists but also and above all surgeons, psychologists, physiatrists, general practitioners, neurologists, nurses, physical and occupational therapists, and so on) to collectively design a shared pain care pathway within which the decision making and the organization of care processes are effectively coordinated by the multi-professional team that builds the care pathway.

From a regulatory standpoint, the absence of indices to quantify and monitor the activities related to pain therapy hampered a clear appreciation of the relevance and the burden of CP within the multiple care services delivered in the context of national health service. The introduction of coding 96 in 2018 [30], which identifies the activities related to pain therapy, is another milestone in the long process toward greater recognition of pain therapy within the national healthcare system. This coding has been waited for a long time by pain specialists and ensures clear documentation of all the activities related to pain therapy at the hospital level. Recognizing a pain code requires healthcare structures monitoring, through an electronic platform, how many hospital units can be devoted to pain therapy, how many beds are available and how many times such coding appears in the discharge letter. Importantly, the availability of such coding may allow specialized pain centers to be accredited for coding 96 thus standing as a reference center for managing CP patients and guiding primary care physicians during patients’ referrals. However, the current implementation of coding 96 is not homogeneous at the national level and no integration among medical disciplines has been attained so far. As a result, patients with CP do not currently benefit from the innovative premises and goals of such coding.

## 4. Healthcare Providers’ Education and Patient Engagement as Pillars to Ensure a Sustainable and Long-Lived Chronic Pain Management

CP has been regarded as a marginalized issue within medical education. It has been suggested that such marginalization may stem from traditional care models which acknowledge pain only as a symptom rather than a disease. In addition, for a long-time teaching pain medicine at medical schools has been missing. As a result, more than half of primary care physicians report discomfort managing CP and rate their residency training in CP management as insufficient [61]. Therefore, the promotion of the advocated cultural transformation of the approach to CP cannot disregard the relevance of the education of both clinicians and nurses. Thus, there is a pressing need to implement medical pain curriculum and to foster new generations of clinicians specialized in CP care thanks to a greater awareness of the clinical relevance of a biopsychosocial approach to CP management [62]. The importance of pain assessment should be better addressed in educational programs. In a survey conducted among GPs less than one in two respondents acknowledged the use of pain assessment scales in follow-up care as clinically relevant. Ensuring high-quality pain management cannot exclude an adequate pain assessment as its absence may negatively influence the quality of pain care [63].

In Europe, the teaching hours devoted to mandatory pain medicine modules contribute to less than 1% of the minimum total training time throughout an undergraduate medical course. In France, almost 90% of the medical schools provide students with dedicated pain courses, while in Germany, since 2012, CP pain modules have been implemented into the medical school curriculum [64]. In Italy, although pain therapy is included in the core undergraduate medical curriculum (not more than two university educational credits), undergraduate physicians still lack effective training that is also frequently absent in post-graduate courses except for the anesthesia and intensive care specialty. In addition, to achieve fully multidisciplinary CP care it is paramount that not only anesthesiologists, whose postgraduate training encompasses specific pain modules, but also the other specialists daily engaged in CP patients’ care could be educated on adequate pain management. To this end, the previously mentioned care pathways may be best suited as an educational resource for early career clinicians or as a part of the curriculum in educational settings [58].

Given the rising future healthcare cost projections, the opportunity to improve health outcomes through patient education and self-management programs should be pursued [65,66]. Patient-oriented interventions should aim at fostering the value of being sufficiently informed: The more clearly a disease is understood by the patient, the more likely it is that an individual will be comfortable with the prescribed therapy, adhere to necessary regimens, and be able to do so safely outside a medical facility.

## 5. Conclusions

The right not to suffer is still disregarded in Italy and the current access to adequate pain care reveals significant shortcomings. Poor disease awareness, late diagnosis, inadequate pain management, and significant inhomogeneity of the delivery of pain care to demand urgent interventions at multiple levels including scientific societies, patients’ associations, and policymakers. To date, recognizing pain as a disease and placing efforts towards full implementation of the ICD-11 coding may increase public awareness of the clinical and social relevance of effective pain care as well as the importance of including pain relief within a continuum of care strategy for CP within the National Chronicity Plan, a health care plan that dictates lines of address on pathologies with a chronic course, which the single regions of Italy must implement on own territory, in consideration of the services and resources available.

Acknowledging pain as a biopsychosocial issue and its impact on multiple domains should set the stage for the development of new thinking on pain management goals and reconsider the current focus on pain intensity. Thus, besides pain intensity, clinicians should be focusing on changes to core domains relevant to the management of CP including pain quality, physical and emotional functioning, improvement and satisfaction with treatment, and therapy tolerability.

A rational approach to CP should involve a model within an integrated continuum of services (primary, secondary, and tertiary) within which multiple professionals cooperate to ensure the enforceability of the right not to suffer. The complexity of CP pathophysiology and the plethora of multidimensional issues patients with CP are suffering from requires the contribution of multidisciplinary teams with the emerging figure of pain medicine specialist who has advanced CP care expertise from diagnosis to therapy prescription and should allay with primary and non-pain specialists to attain meaningful patients’ outcomes. To this end, the improvement of education in all medical specialties, by promoting specific under and post-graduate courses, and the development of a common language and diagnostic criteria will ensure a shared multidisciplinary approach to CP. Finally, establishing a clear management pathway may hold great potential to optimize pain care. Interventions aiding in appropriate referral and favoring care coordination could ensure that people with CP are taken care of by the right physician at the right time. Early access to pain care, by minimizing a potential late diagnosis and accomplishing an effective treatment thus avoiding costly complications, may improve patients’ QoL and the sustainability of healthcare systems.

In conclusion, we hope that our perspective serves to foster debate around the strategies needed to be implemented at the national level to ensure full enforceability of the right not to suffer via the development of effective and widespread multidisciplinary care of patients with CP.

## Figures and Tables

**Table 1 ijerph-20-00551-t001:** Snapshot of the main features and clinical implications of Law 38/2010. Elaborated from the text in [12].

Pain is an issue that should be addressed systematically, at every stage of the disease journey and in every care setting
The right not to suffer is a regional * fulfillment for access to Essential Levels of Care
There are minimum criteria and organizational premises needed to authorize pain care palliative facilities and pain therapy networks in both adult and pediatric patients
Professional figures with expertise in palliative care and pain therapy exist including general practitioners, specialists in anesthesia and intensive care, geriatrics, neurology, oncology, radiotherapy, pediatrics, clinicians with at least three years of experience in palliative care and pain therapy, nurses, psychologists, social assistants, and any other healthcare professionals who are considered essential
Continuity of care between care providers and care settings is essential to receive adequate access to pain care
Pain is a medical issue on its own and pain assessment should be included in medical records
The right to health as established by Article 32 of the Italian Constitution is reinforced as an obligation to the development of a system endowed by mutual support for all citizens particularly frail individuals
Care models for chronic pain are redefined by developing dedicated care networks, placing emphasis on previously underserved populations (e.g., children), and simplifying the procedures to access drugs for pain therapy

* Italy encompasses 20 regions which are the first-level constituent entities of the Italian Republic. The administration of the healthcare system is the responsibility of the regional government.

**Table 2 ijerph-20-00551-t002:** Social Manifesto against pain [23]. Reproduced with the permission of SIAARTI.

1	**Access to pain therapy: An enforceable right** Access to non-cancer chronic pain care should be regarded as unalienable and enforceable by citizens, convenient for the whole society, guaranteed by the presence of specific and committed healthcare resources
2	**A diffuse and homogeneous right** There is a need for a comprehensive, homogeneous, and territorial diffusion of pain therapy across the country
3	**A fully proportionate right** There is a need to collect over time data to define the number of pain center therapy necessary across the country based on the number of inhabitants to address the patient’s needs and to keep updated the census of pain therapy centers in relation to the number of patients treated and followed up.
4	**A right without waiting** The pain therapy centers should be fully equipped and functional to provide the citizens with homogeneous access to therapy and follow-up across the country by minimizing the waiting list and ensuring an adequate care pathway for patients with non-cancer chronic pain
5	**A right for the major frailties** There is a need that all the most vulnerable and frail subpopulations can experience full, rapid, facilitated and continuo access to pain therapy centers
6	**A right ensured by a specific multidisciplinary expertise** All the healthcare professionals involved in pain treatment should be included in care pathways and engaged in high-quality, evidence-based education programs
7	**A right research-based** Pain research should apply to the funding for innovative drugs development and national agencies should support and promote the independent research
8	**A right supported by digital technologies** High-tech solutions should be developed to achieve effective remote monitoring, care, and consulting of patients suffering from chronic pain
9	**A monitored right** Institutions at both central and regional levels along with scientific societies and patients’ associations should acquire tools to monitor over time and accurately the level of implementation of law 38/2010 and the subsequent provisions
10	**A communicated right** All media should be engaged and committed to delivering adequate information about pain therapy with the helpful support of the patients’ associations and citizen organizations

## Data Availability

Not applicable.

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
