# Peer review of "Raising Awareness on the Clinical and Social Relevance of Adequate Chronic Pain Care"

_ijerph, 2022, doi:10.3390/ijerph20010551_

Round 1

Reviewer 1 Report (Previous Reviewer 1)

I think the revisions took care of reviwer comments. The MS now seems to be fit to foster the debate on implementation strategies for multidisciplinary care of pain patients in Italy and elsewhere.

Reviewer 2 Report (Previous Reviewer 2)

None to the authors.

This manuscript is a resubmission of an earlier submission. The following is a list of the peer review reports and author responses from that submission.

Round 1

Reviewer 1 Report

This manuscript addresses a topic of high current interest in the field of public health: implementation of chronic pain management in a European country (Italy). The Law 38 in Italy in 2010 raised a lot of attention in Europe and expectations for its implementation were high, with Italy possibly serving as a model country in the future. It is highly relevant to learn, if this is not the case, what went wrong and how the mistakes may be corrected or avoided. So the MS is clearly of international relevance although its content is specifically about Italy. The current MS, however, falls short of fulfilling such expectations.

Major issues:

1.     Such a statement should be issued by more than one medical society. On p. 2 the authors mention that SIAARTI has cooperated with other scientific societies and patient groups on this topic. The author addresses suggest that this may be a multidisciplinary panel, but liaisons with other societies – if they exist – are not listed. I think there is an Italian Pain Society and a Headache Society, and neurologists and psychologists are also very active in pain medicine.

2.     P. 8 mentions a joint call to action by many stakeholders. Is this completely covered by Vittori et al. 2021 (ref 20)? Then what is the news value of the current MS?

3.     The text is often quite vague and somewhat repetitive. The facts that this analysis is based on should be spelled out clearly and possibly summarized in a Table. The text can be shortened overall by 30%.

4.     Multimodal treatment is mentioned but specific recommendations are mainly about opioids, which play only a minor role in chronic pain management. These aspects need to be more balanced, including specific recommendations non-opioid medications, physiotherapy and psychotherapy availability.

5.     Two potentially interesting legislative actions in Italy are briefly mentioned (coding 96, national chronicity plan). These need to be explained to readers from outside Italy.

6.     Are there any national guidelines on pain management in Italy, and what role do they play for the agenda?

7.     Which pain medicine curricula would the authors recommend to be implemented in order to achieve their goals?

Minor issues:

1.     There is a typo in address 10

2.     Ref 21 and 22 appear to be identical

3.     On p. 7 ref 62 is quoted for pain medicine education in Germany; I think this should be ref 63

Reviewer 2 Report

The authors are apparently members of the SIAARTI board and they discuss the roadblocks to chronic pain management.

Unfortunately, I don't see any new or valuable information in this paper.